# Genome-Wide Analysis of microRNAs and Their Target Genes in Dongxiang Wild Rice (*Oryza rufipogon* Griff.) Responding to Salt Stress

**DOI:** 10.3390/ijms24044069

**Published:** 2023-02-17

**Authors:** Yong Chen, Wanling Yang, Rifang Gao, Yaling Chen, Yi Zhou, Jiankun Xie, Fantao Zhang

**Affiliations:** 1College of Life Sciences, Jiangxi Normal University, Nanchang 330022, China; 2State Key Laboratory of Crop Gene Exploration and Utilization in Southwest China, Sichuan Agricultural University, Chengdu 611130, China

**Keywords:** wild rice, miRNA, target gene, salt tolerance, genetic resource

## Abstract

Rice (*Oryza sativa*) is a staple food for more than half of the world’s population, and its production is critical for global food security. Moreover, rice yield decreases when exposed to abiotic stresses, such as salinity, which is one of the most detrimental factors for rice production. According to recent trends, as global temperatures continue to rise due to climate change, more rice fields may become saltier. Dongxiang wild rice (*Oryza rufipogon* Griff., DXWR) is a progenitor of cultivated rice and has a high tolerance to salt stress, making it useful for studying the regulatory mechanisms of salt stress tolerance. However, the regulatory mechanism of miRNA-mediated salt stress response in DXWR remains unclear. In this study, miRNA sequencing was performed to identify miRNAs and their putative target genes in response to salt stress in order to better understand the roles of miRNAs in DXWR salt stress tolerance. A total of 874 known and 476 novel miRNAs were identified, and the expression levels of 164 miRNAs were found to be significantly altered under salt stress. The stem-loop quantitative real-time PCR (qRT-PCR) expression levels of randomly selected miRNAs were largely consistent with the miRNA sequencing results, suggesting that the sequencing results were reliable. The gene ontology (GO) analysis indicated that the predicted target genes of salt-responsive miRNAs were involved in diverse biological pathways of stress tolerance. This study contributes to our understanding of DXWR salt tolerance mechanisms regulated by miRNAs and may ultimately improve salt tolerance in cultivated rice breeding using genetic methods in the future.

## 1. Introduction

Rice (*Oryza sativa*) is a cereal crop that feeds more than half of the world’s population, especially in Asia, where approximately 80% of global rice is cultivated and consumed [1,2]. It is predicted that food production will need to increase by about 70% by 2050 to maintain sufficient food levels for the population [3]. As observed in paddy field crops, rice production decreases when subjected to abiotic stresses such as water deficiency or submergence, low or high temperatures, and high salinity [4]. After drought, salinity is the second most prevalent soil problem in rice-growing countries [5]. Approximately 30% of the world’s rice lands contain too much salt to allow normal rice cultivation, and the rice yield is reduced by 68% when cultivated on such moderately salt-affected soils [6]. Even worse, as the global temperatures continue to rise, more soil in semiarid regions will be salinized by irrigation with saline water due to water scarcity and rising sea levels [7]. Therefore, salinity is considered one of the greatest environmental threats to rice production worldwide, and attaining rice cultivars that are tolerant to high salt is of the utmost necessity for the agricultural sector.

MicroRNAs (miRNAs) are short non-coding RNA molecules (20–24 nt) that regulate the expression of protein-coding genes at the post-transcriptional level by cleaving their target genes [8,9]. Almost all pathways in the eukaryotic gene regulation system are directly or indirectly regulated by miRNAs [10]. In the past few decades, a vast number of miRNAs has been identified in various plant species using high-throughput sequencing technology [11]. Meanwhile, many stress-specific miRNAs have been identified under different biotic and abiotic stress conditions such as high sanity [12], drought [13], cold [14], nutrient deficiency [15], and infection [16], suggesting that miRNAs play very important roles in various stress responses in plant species. However, previous related studies have predominantly focused on the important model plants and agricultural crops. More research needs to be done to further elucidate miRNA functions among more plant species.

Common wild rice (*Oryza rufipogon*) is thought to be the progenitor of cultivated rice, and 30–40% of its genetic variation was estimated to be lost during the domestication process [17]. Dongxiang wild rice (*Oryza rufipogon* Griff., DXWR) is the northernmost (28^o^14′ N) common wild rice ever found in the world, and has been found to tolerate various abiotic stresses [18,19]. Meanwhile, the previous studies revealed that DXWR has higher tolerance to salt stress than cultivated rice, making it a unique gene pool for identifying more precious salt stress response genes. Although some miRNAs have been identified in DXWR by high-throughput sequencing and a bioinformatics approach, the salt-responsive miRNAs have been little identified and characterized in DXWR, and the expression pattern of miRNAs under the salt stress of DXWR remains unclear.

In this study, a comprehensive view of known and novel miRNAs and their expression patterns under salt stress is characterized using high-throughput sequencing technology. A set of differential expression miRNAs were verified using stem-loop quantitative real-time PCR (qRT-PCR), and the target genes of salt stress-responsive miRNAs were predicted and characterized. This study helps to clarify the response of miRNAs and their target genes to salt stress, to explore the miRNA-regulated mechanism of salt stress tolerance in DXWR, and finally to help genetically improve salt stress tolerance in cultivated rice in the future.

## 2. Results

### 2.1. Overview of sRNA Library Data Sets

To investigate the possible miRNAs involved in the salt stress response in DXWR, six small RNA libraries from the control and salt treatment groups were constructed and subjected to high-throughput sequencing, named DY-CK1, DY-CK2, DY-CK3, DY-S1, DY-S2, and DY-S3. The DY-CK (1–3) libraries were three biological replicates of DXWR under normal conditions, and the DY-S (1–3) libraries were three biological replicates of DXWR after salt treatment. In total, 96.34 million raw short reads were obtained from the six libraries, with 16.06 million raw reads per library on average. After filtering out low-quality data, 3′ joint contamination data, and sequences with a length less than 18 nt or greater than 25 nt, a total of 43.88 million clean reads were obtained, with a mean of 7.31 million clean reads per library (Table 1). Meanwhile, the Q30 (sequencing error rate  <  0.1%) scores of all libraries ranged from 94.05% to 95.18%, with an average of 94.54%, indicating that the sequence data were of reliable quality.

To further analyze the validity of the sequence data, a statistical analysis on the length distribution of total and unique sRNAs was performed on filtered datasets. Of all the total sRNAs, 24 nt-sRNAs were the most abundant, accounting for an average of 24.5% and 21.8% of the DY-CK and DY-S libraries, respectively (Figure 1A). Among the unique sRNAs, 24 nt-sRNAs were the most frequent, accounting for an average of 39.6% and 38.6% in the DY-CK and DY-S libraries, respectively, which was consistent with the typical size of miRNAs from Dicer-derived products (Figure 1B).

### 2.2. Identification of Known and Novel miRNAs in DXWR

To identify the known miRNAs of DXWR under normal and salt treated conditions, the unique clean reads were blasted to the miRbase database for comparison with the currently known plant precursor or mature miRNA sequences. In total, 712 pre-miRNAs corresponding to 874 known unique mature miRNAs were identified as homologues of known miRNAs from the other plants, such as *Arabidopsis thaliana*, *Cynara cardunculus*, *Glycine max*, *Medicago truncatula*, and *Triticum aestivum* (Appendix A). Among the 874 known miRNAs, only 63 miRNAs showed high expression levels (reads greater than the average copy of the data set), and osa-miR168a-5p, osa-miR166a-3p, osa-miR1425-5p, osa-miR168a-3p_L-3, and osa-miR396e-5p were identified as the most abundantly expressed conserved miRNAs in DXWR. A majority of the known miRNAs showed middle (520 miRNAs with reads greater than 10 but below the average copy of the data set) to low (291 miRNAs with reads less than 10) expressional levels (Appendix A). Among these known miRNAs, 523 belong to 69 families (Appendix A), whereas the families of the other 351 miRNAs were unknown. The three largest families were miR812 (54 miRNA members), miRNA166 (27 members), and miR814 (26 members), whereas most families contained less than 10 members (Appendix A and Appendix A).

In addition, the unmapped sequences were compared with the rice genome, and mapped sequences that fulfilled the criteria for annotation of plant miRNAs were identified as novel miRNAs. Finally, a total of 476 novel miRNAs were identified from 528 pre-miRNAs (Appendix A). A majority (66.0%) of the identified novel miRNAs showed low expressional levels, 162 (34.0%) of the novel miRNAs showed middle abundance, and no novel miRNAs exhibited high abundance (Appendix A). Among the novel miRNAs, the first nucleotides of 5′ were biased toward A (adenine) (59.0%) and U (uracil) (20.6%) (Appendix A). These pre-miRNAs range in length from 56 nt to 255 nt with an average length of 149 nt, which is consistent with the general length of pre-miRNAs. The CG percentages (CG%) of these novel pre-miRNAs range from 18.5 to 78.9%, and their minimal folding free energy index (MFEI) ranges from 0.9 to 2.3 with an average of 1.4 (Appendix A).

### 2.3. Differential Expression Analysis of miRNAs in DXWR under Salt Stress Condition

To identify differentially expressed miRNAs (DEMs) that responded to salt stress, the expression levels of all miRNAs in the DY-CK and DY-S libraries were normalized and analyzed. Intriguingly, 256 and 91 miRNAs were specifically expressed in the normal and salt stress conditions, respectively (Appendix A), implying that the specifically expressed miRNAs under the normal condition may play negative roles in the salt response, whereas those under the salt stress condition may play positive roles in the salt response in DXWR. Meanwhile, of the 1,350 (874 known and 476 novel) identified miRNAs, the expressions of 164 miRNAs, including 139 known and 25 novel miRNAs, were significantly altered (*p* < 0.05), and over half of the DEMs (99 out of 164) were downregulated (Appendix A). The expression levels of osa-miR399j_R-1 and osa-miR2106_R+1 significantly decreased and increased (−3.91 and 4.31 log_2_FC; FC means fold change), respectively (Figure 2). After similar sequences of DEMs were assigned to their family, most DEMs within their miRNA family displayed similar expression patterns, such as the family numbers of miR164 and miR166 being significantly downregulated under salt stress. Among the known DEMs, the miR166 family had the highest numbers (12), followed by the miR169_1 (9) and miR171_1 (7) (Appendix A).

To validate the high-throughput sequencing data and expression patterns of miRNAs, ten DEMs that show significant expression changes after salt treatment were randomly selected. The stem-loop qRT-PCR results showed that osa-miR5072_L-4, ath-miR8175_L-2, ptc-miR6478_1ss21GA, and osa-miR2106_R+1 were upregulated and osa-miR167d-3p, osa-miR166h-5p, osa-miR171g-p5, osa-miR1850.1, osa-miR166j-5p, and PC-5p-16439_287 were downregulated. The expression trends of these miRNAs in control libraries relative to those in salt-treated libraries detected by small RNA sequencing were basically consistent with those detected by stem-loop qRT-PCR (Figure 3, Appendix A), suggesting that the miRNA sequencing data results were credible.

### 2.4. Prediction and Functional Annotation of the Known and Novel DEMs Targets

Targets of known and novel DEMs were predicted using RNAplex software, and a total of 2018 transcripts from 1774 genes were identified (Appendix A). Among them, 1900 transcripts from 1666 genes were predicted to be the targets of 127 known DEMs, and 143 transcripts from 133 genes were predicted to be the targets of 21 novel DEMs (Appendix A). Of the 1,774 target genes, 115 genes are transcription factors (TFs), such as *HSFC1A*, *HSFC2A*, *OsTIFY1A*, *OsERF101*, *OsbHLH113*, and *R2R3-MYB* (Appendix A), which are DNA binding transcription factors and the regulation of stress response. The other miRNA target genes, including peptidase (such as serine carboxypeptidase-like, endoplasmic reticulum metallopeptidase, and carboxyl-terminal-processing peptidase), synthase (such as 3-ketoacyl-CoA synthase, noroxomaritidine synthase, and starch synthase), and transferase (such as acetyltransferase, O-fucosyltransferase, and nicotinate phosphoribosyltransferase) genes were involved in plant growth, development, and abiotic stress responses. These results indicated that miRNAs may play an important role in different biological processes under salt stress in DXWR.

To further analyze the specific biological functions of DXWR miRNAs under salt stress, a Gene ontology (GO) analysis was performed to explore the biological functions of the 1774 target genes for known and novel DEMs; the results indicated that the target genes were mainly annotated into 50 GO terms, which were most related to the biological process category (25 terms), followed by the cellular component (15 terms) and molecular function (10 terms) categories (Figure 4). In the classification of the cellular component category, the GO terms including nucleus (384 genes), plasma membrane (237 genes), cytoplasm (180 genes), and integral component of membrane (175 genes) have the top four number of genes; in the molecular function classification, GO terms of protein binding (176 genes) and molecular function (130 genes) have the most target genes; in the biological process category, biological process (144 genes) and regulation of transcription (114 genes) have the most gene numbers, and in this category, 56 target genes were annotated in response to stress, of which 26, 17, and 13 target genes were annotated in response to salt, oxidative, and osmotic stress, respectively (Appendix A). From the GO terms, we found that many processes, such as responses to stress (GO:0006950), abscisic acid transport (GO:0080168), auxin transport (GO:0060918), and rRNA binding (GO:0019843) were prominently down-regulated and that the processes of responses to DNA demethylation (GO:0080111), water homeostasis (GO:0030104), secondary growth (GO:0080117), and histone acetylation (GO:0016573) were up-regulated (Appendix A).

Furthermore, our previous study revealed an expression profile of genes in DXWR under salt stress: 743 genes were downregulated, while 892 were upregulated in both roots and leaves [20]. Among these genes, 81 genes coexisted in the target genes predicted in this study, and 41 miRNA–mRNA pairs showed opposite expression patterns in DXWR under salt stress (Table 2). Among them, nine genes have been named. Meanwhile, five of the nine genes have been well studied and proved to be associated with various abiotic stresses, including *OsERF101* (Os04g0398000), *OsSPX-MFS1* (Os04g0573000), *OsKOS1* (Os06g0569500), *OsPDK1* (Os07g0637300), and *OsRab16A* (Os11g0454300). The upstream miRNAs of the five genes were downregulated in this study, and when we measured the expression levels of the five genes in DXWR under salt stress, all were upregulated, consistent with the previous results and negatively correlating with the miRNA expression (Figure 5). Therefore, these miRNAs and their putative targets could be potential salt stress-associated regulators that deserve further investigation.

## 3. Discussion

After drought, salinity is the second most common soil problem in rice-cultivating countries and has become a serious obstacle to improving global rice production [21]. More than 50% of arable lands may be lost to serious salinization by the year 2050, making it difficult to secure rice production and exacerbating food shortages [22]. Although rice salt tolerance has been improved by molecular biological techniques in recent years [23,24], surprisingly little is known about the mechanistic basis of the salt response. Wild rice is characterized by its excellent agronomic traits and tolerance to biotic and abiotic stresses [25]. During the domestication process, genetic diversity is rapidly lost, resulting in the loss of genes related to useful agronomic traits [26]. As a progenitor of cultivated rice (*Oryza sativa* L.), Dongxiang wild rice (*Oryza rufipogon* Griff., DXWR) has survived natural selection, and its gene diversity and tolerance to abiotic and biotic stress has been lost in cultivated rice [20]. Therefore, the biological mechanism underlying DXWR tolerance to stress requires further study.

MiRNAs are universally present in plants and mediate gene expression through target cleavage or translational repression [11]. Numerous stress-responsive miRNAs have been identified using sequencing technology since the first study reported miRNAs involved in the plant stress response in *Arabidopsis* [27]. Interestingly, as some miRNAs manage the metabolic pathway network in response to biotic and abiotic stress, it is important to investigate the roles of miRNAs in DXWR salt tolerance [28]. In this study, 874 known and 476 novel miRNAs were identified using high-throughput sequencing technology on a genome-wide scale in DXWR (Appendix A), of which 99 miRNAs were significantly downregulated and 65 were upregulated under salt stress (Appendix A). Among the DEMs, miR156 [29], miR159 [30], miR160 [31], miR164 [32], miR166 [33], miR167 [34], miR169 [35], miR171 [36], miR172 [37], miR399 [38], miR444 [39], and miR827 [40] have been reported to mediate stress responses in plants. Additionally, we identified known and novel miRNAs in DXWR under drought stress and found that the expression levels of miR160, miR164, miR166, miR167, miR172, miR444, miR810, miR5072, and miR1846 were significantly different [19,41]. Moreover, miR160, miR166, miR167, miR172, and miR5072 expression patterns coincided with miRNA expression in this study, suggesting that these miRNAs may retain the same properties when DXWR is subjected to drought and salinity stresses.

In addition, 2018 transcripts were predicted as candidate target genes for the salt-responsive DEMs in DXWR. The majority of these transcripts were predicted as functional genes encoding TFs, peptidases, synthases, transporters, or transferases. TFs are considered to have the greatest effects on plant salinity tolerance because these key regulators can regulate the expression levels of a range of salinity tolerance genes [42]. The bZIP family is one of the largest TF families in higher plants. Until now, many bZIP genes involved in salt stress responses have been characterized in various plant species. Gong et al. identified 48 bZIP genes in the genome of sugar beet (*Beta vulgaris* L.) and analyzed their biological functions and response patterns to salt stress [43]. Chai et al. revealed that the overexpression of a soybean bZIP gene *GmbZIP152* can enhance the tolerance to salt, drought, and heavy metal stresses in *Arabidopsis* [44]. In this study, four bZIP genes (*OsbZIP17*, *OsbZIP27*, *OsbZIP81*, and *OsbZIP85*) were predicted to be targeted by the salt-responsive DEMs. Intriguingly, Liu et al. found that *OsbZIP81* can regulate jasmonic acid (JA) levels by targeting the genes in the JA signaling and metabolism pathways in rice [45]. Many studies have revealed that JA can mediate the effect of abiotic stresses and help plants to acclimatize under stress conditions [46]. Thus, *OsbZIP81* and its corresponding miRNAs could be potential salt-associated regulators that deserve further investigation. Meanwhile, an increasing number of studies have shown that transferase genes help plants to respond and adapt to abiotic stresses. Sun et al. reported that ectopic expression of the *Arabidopsis* glycosyltransferase UGT85A5 gene can enhance salt stress tolerance in the plants of tobacco [47]. Duan et al. identified a total of 189 UDP-glycosyltransferase genes in the *Melilotus albus* genome and revealed their vital roles in abiotic stress responses [48]. In this study, 59 transferase genes were predicted to be targeted by the salt-responsive DEMs. Certainly, further investigation is required to confirm, if any, the roles of these transferase genes and their corresponding miRNAs involved in the salt tolerance of DXWR.

Meanwhile, the expression patterns of miRNAs and target genes can be used as indicators to determine the function of miRNAs, since the miRNAs and target genes were usually oppositely expressed. Previously, we analyzed the transcriptome profiles of DXWR under salt stress [20]. In this study, the expression patterns of miRNAs and target genes were evaluated by a combination analysis of miRNAs and transcriptome profiles. Finally, we screened out five stress-related genes for further investigation based on functional annotation, i.e., *OsERF101*, *OsSPX-MFS1*, *OsKOS1*, *OsPDK1*, and *OsRab16A*. *OsERF101* is an ethylene-responsive factor that is mainly expressed in rice reproductive tissues; *OsERF101*-overexpression plants were more tolerant to osmotic stress than wild-type plants, and when subjected to drought stress in the reproductive stage, transgenic plants had higher survival and seed setting rates [49]. *OsSPX-MFS1* belongs to the *SPX-MFS* family and is mainly expressed in shoots; *OsSPX-MFS1* mutant (*mfs1*) or overexpression of the upstream regulator miR827 impairs phosphate homeostasis [50]. *OsKOS1* is an *ent*-kaurene oxidase-like protein, and its expression is induced by UV irradiation and is likely to participate in phytoalexin biosynthesis [51], whereas phytoalexin is a compound that regulates the interaction between parasites and the host plant [52]. *OsPDK1* can positively regulate basal disease resistance in rice [53]. *OsRab16A* is induced when plants are subjected to stress, and overexpression of *Rab16A* increases plant salt tolerance [54,55]. The qRT-PCR analysis results showed that the expression levels of these five genes were all upregulated under salt stress in DXWR, negatively correlating with their upstream miRNAs’ expression. Most notably, the expression level of *OsRab16A* (*Os11g0454300*) was the most significantly changed among the five genes, which further implies the role of this gene in conferring salt stress tolerance in plants.

Furthermore, GO enrichment analyses of target genes can help us to understand the functions of miRNAs more effectively. Our functional prediction according to GO categories showed that the target genes were significantly enriched in molecular functions, including DNA-binding TF activity, TF binding, L-alanine transmembrane transporter activity, arginine transmembrane transporter activity, and gamma-aminobutyric acid transmembrane transporter activity. As mentioned above, genes involved in DNA-binding TF activity and TF binding have shown vital roles in regulating stress tolerance in plants [42,43,44,45]. Meanwhile, according to Zhou et al. [56], miRNAs can also be involved in stress tolerance by regulating target genes that control transmembrane proteins. Thus, our findings are in agreement with the results of previous studies. In addition, plant hormones and their signal transductions play important roles in response to various abiotic stresses [57]. Based on the annotation of target genes, we identified that 14, 3, 2, 2, and 1 target genes of salt-responsive DEMs were associated with ethylene, auxin, abscisic acid, gibberellin, and cytokinin, respectively. Therefore, these results provide an abundant resource of candidate miRNAs and target genes associated with salt tolerance and their enriched regulatory networks in plants.

In the past two decades, some stress-responsive miRNAs have shown potential in stress tolerance improvement of various plant species. Moreover, miRNA modulation has been successfully developed to improve plant stress tolerance using many available biotechnological tools, such as overexpression, RNAi, and CRISPR Cas9 systems. For instance, transgenic tobacco plants expressing Zm-miR156c exhibit enhanced drought and salt stress tolerance [58]. The overexpression of sha-miR319d increases chilling and heat tolerance in tomato plants [59]. The overexpression of miR408 leads to enhanced resistance against cold, salt, and oxidative stresses in *Arabidopsis* [60]. The deletion of miR169a by CRISPR/Cas9 increases drought stress tolerance in *Arabidopsis* [61]. Therefore, we believe that the identified salt-responsive miRNAs from DXWR could provide a basis for developing salt stress-tolerant rice varieties through molecular design breeding in the future.

## 4. Materials and Methods

### 4.1. Plant Materials, Culture, and Sample Collection

The seeds of Dongxiang wild rice (*Oryza rufipogon* Griff., DXWR) were stored in our laboratory of Jiangxi Normal University. Seed germination and cultivation were carried out in accordance with the previous study [19]. Salt treatment was carried out when the rice plants were at the four-leaf stage. After two days of treatment with and without 200 mM salt, seedlings were collected and immediately frozen with liquid nitrogen.

### 4.2. Small RNA Library Construction and Deep Sequencing

To explore the regulatory mechanisms of miRNAs in response to salt stress in DXWR, we constructed two sample groups, namely, the salt-treated group (DY-S) and normal control group (DY-CK). Each sample group contains three biological replicates, named DY-CK1, DY-CK2, DY-CK3, DY-S1, DY-S2, and DY-S3. Library preparation and sequencing-related experiments were performed in accordance with the standard procedure provided by Illumina (San Diego, CA, USA). The TruSeq Small RNA Sample Prep Kit (Illumina, San Diego, CA, USA) was used for the preparation of the sequencing library. Then, the prepared libraries were sequenced by the Illumina Hiseq2000/2500 sequencing system (Illumina, San Diego, CA, USA) with a single-end 50 bp read length. The library construction and deep-sequencing analysis were performed by Lianchuan BioTech Co., Ltd. (Hangzhou, Zhejiang, China).

### 4.3. Sequencing Data Analysis and Identification of Known and Novel miRNAs

Raw sequencing reads were analyzed by the ACGT101-miR program (LC Sciences, Houston, TX, USA). After removing the 3′ adapters and junk sequences, the remaining sequences with a length of 18–25 nt were aligned to mRNA (http://rapdb.dna.affrc.go.jp/download/irgsp1.html, accessed on 3 September 2022), Rfam (http://rfam.janelia.org, accessed on 3 September 2022), and repeat (http://www.girinst.org/repbase, accessed on 3 September 2022) databases to remove the matched sequences, respectively. The remaining valid reads were blasted against miRbase (http://www.mirbase.org/, accessed on 3 September 2022) [62] to identify the known miRNAs. The remaining unmapped sequences were compared with the rice genome (http://rapdb.dna.affrc.go.jp/download/irgsp1.html, accessed on 3 September 2022), and mapped sequences that fulfilled the criteria for the annotation of plant miRNAs were identified as novel miRNAs [63]. The *p* value of the Student’s *t*-test was used to analyze the DEMs based on normalized deep-sequencing counts; the *p*-value ≤ 0.05 was set as the significance threshold of DEMs in this test.

### 4.4. Verification of Sequencing Data

To verify the reliability of the sequencing data, miRNAs were randomly selected from DEMs for qRT-PCR analysis. RNAs were extracted using a TRIzol reagent (Sangon Biotech Co., Ltd., Shanghai, China), following the manufacturer’s instructions. MiRNA reverse transcription was performed using the miRNA 1st Strand cDNA Synthesis Kit (by stem-loop) (Vazyme, Nanjing, China), and mRNA reverse transcription was performed using the PrimeScript™ RT reagent Kit (Takara, Dalian, China). The qRT-PCR experiments were performed using the SYBR Premix Ex Taq II kit (Takara, Dalian, China) on an ABI 7500 Real-Time System (Applied Biosystems, Carlsbad, CA, USA). U6 small nuclear RNA (U6snRNA) and actin genes were used as endogenous controls to normalize the threshold cycle (Ct) values for the miRNAs and mRNAs detected by qRT-PCR, respectively. The relative expression levels of the miRNAs and mRNAs were calculated using the 2^−ΔΔCt^ method [64]. All reactions were repeated three times. The primers used in this study were listed in Appendix A.

### 4.5. Prediction of Target Genes for Salt Stress-Responsive miRNAs

Based on miRNA sequencing, the putative target genes of differentially expressed miRNAs were predicted using RNAplex software 2.5.1 (http://www.tbi.univie.ac.at/RNA/RNAplex.1.html, accessed on 5 September 2022) [65] with a minimum free energy ratio (MFE ratio) cutoff >0.65. The putative target genes were then used for gene ontology (GO) enrichment analysis, which was conducted with a hypergeometric distribution (LC-BIO, Hangzhou, China).

## 5. Conclusions

In summary, we performed small RNA sequencing of DXWR during salt stress and identified 874 known and 476 novel miRNAs. Among these, 65 and 99 miRNAs were significantly upregulated and downregulated, respectively. The predicted target genes of salt-responsive miRNAs were annotated to participate in multiple biological processes. Our findings provide a comprehensive view of miRNA regulation of target genes in DXWR under salt stress, serving as a useful resource for better understanding the biological mechanisms of salt tolerance and for developing tolerant rice breeding practices.

## Figures and Tables

**Figure 1 ijms-24-04069-f001:**
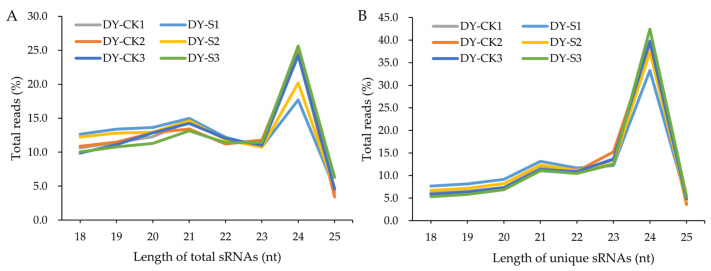
Length distribution and abundance of small RNAs in the libraries of DY-CK and DY-S samples. (**A**) Size distribution of total small RNAs; (**B**) Size distribution of unique small RNAs.

**Figure 2 ijms-24-04069-f002:**
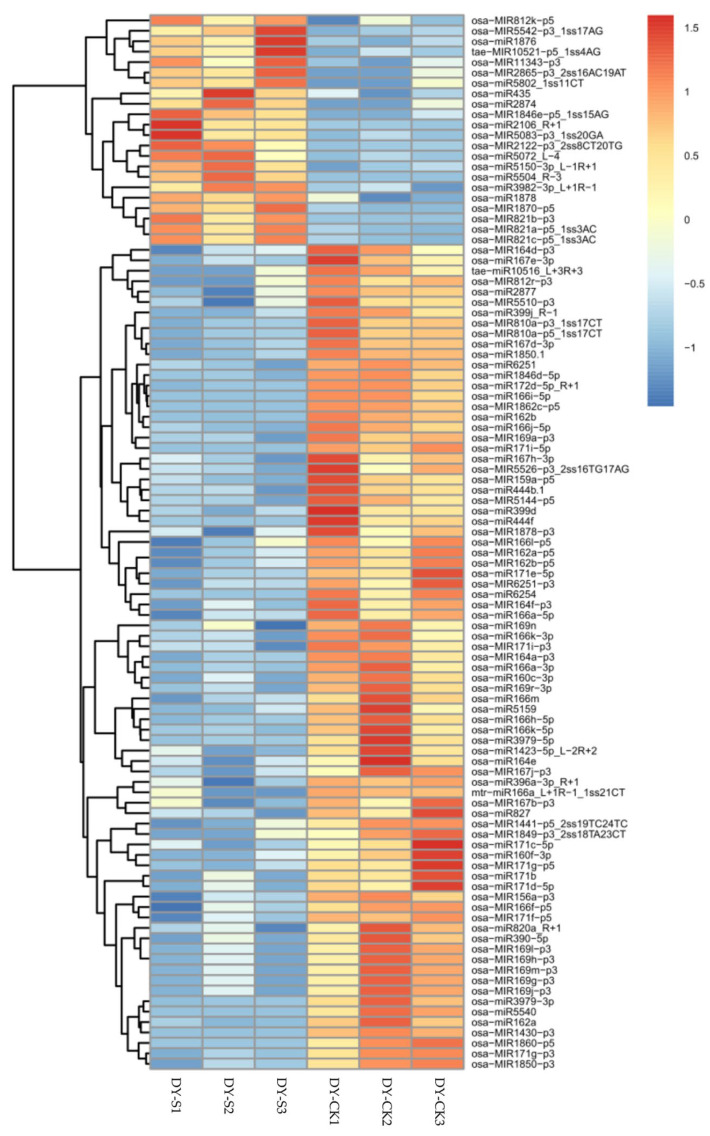
Heat map analysis of differentially expressed miRNAs in six libraries. Up- and downregulated genes were indicated in red and blue, respectively. Color brightness reflected the magnitude of difference.

**Figure 3 ijms-24-04069-f003:**
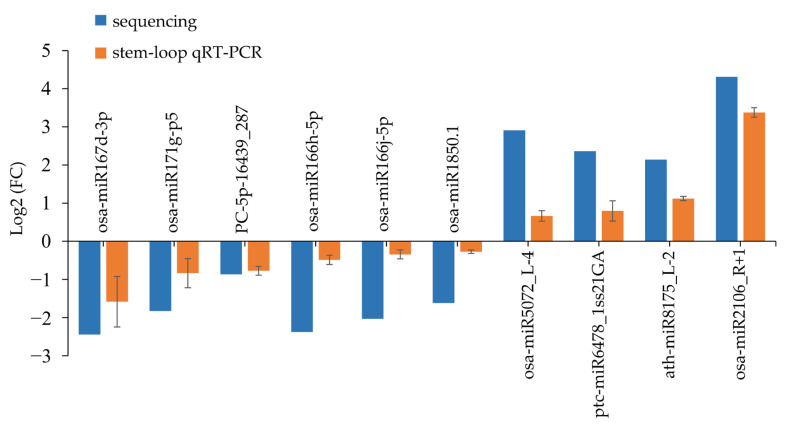
qRT-PCR validation of ten randomly selected salt stress-responsive miRNAs’ relative expression (fold changes of sequencing reads and qRT-PCR) between control and salt-treated library. The blue bars represent the fold change (log2) in control libraries relative to that in salt-treated libraries detected by small RNA sequencing, while the orange bars represent the fold change (log2) in control libraries relative to that in salt-treated libraries detected by stem-loop qRT-PCR (normalized to U6snRNA; n = 3).

**Figure 4 ijms-24-04069-f004:**
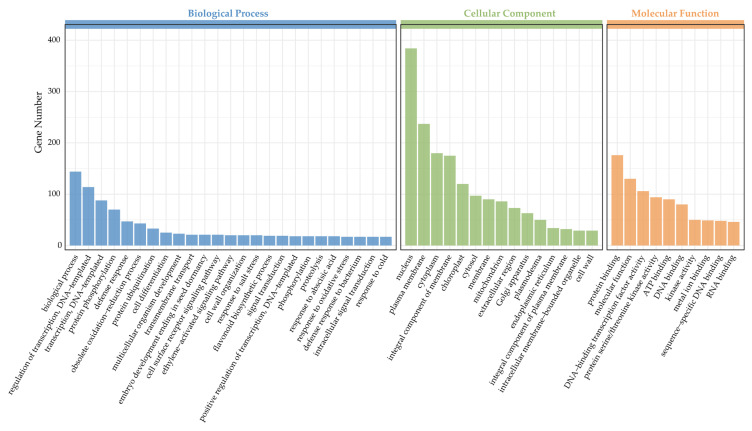
Gene ontology classification of the targeted genes of salt stress-responsive miRNAs.

**Figure 5 ijms-24-04069-f005:**
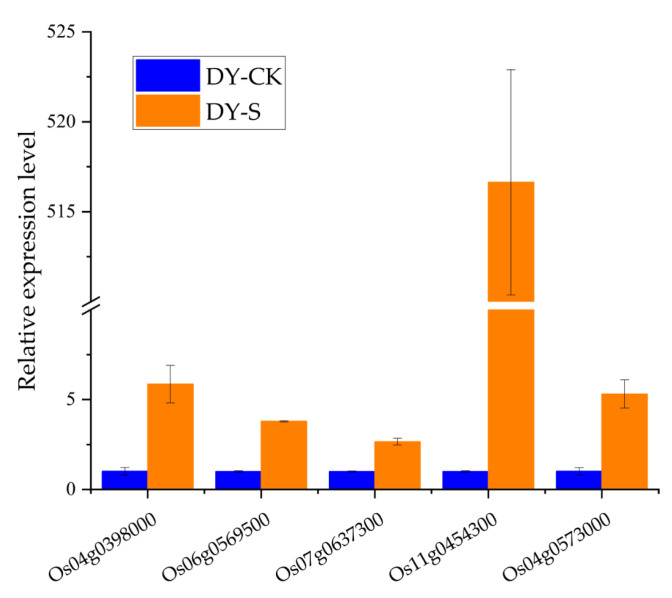
The expression levels of target genes under salt stress detected by qRT-PCR. The blue bars represent the samples under the normal condition, and the orange bars represent the samples under salt stress. The relative expression was calculated by the 2^−ΔΔCt^ method, and the standard deviation was calculated with three biological repeats.

**Table 1 ijms-24-04069-t001:** Summary of small RNA sequencing in the DXWR at control and salt stress conditions.

**Types**	**DY-CK1**		**DY-CK2**		**DY-CK3**	
	**Total**	**Unique**	**Total**	**Unique**	**Total**	**Unique**
Raw reads	10,618,693	2,604,513	16,460,121	3,672,322	22,194,190	5,038,513
3′adaptor & length filter	5,585,638	1,009,906	9,545,474	1,324,469	11,078,492	1,954,931
Junk reads	20,571	14,159	28,759	20,791	50,332	31,211
Clean reads	5,012,484	995,747	6,885,888	1,303,678	11,065,366	1,923,720
Rfam	637,243	14,603	859,520	15,142	1,357,192	24,867
mRNA	590,743	14,426	667,359	20,178	1,335,628	35,142
Repeats	9114	205	11,931	236	18,131	327
valid reads	3,817,501	1,552,466	5,396,359	2,292,925	8,450,225	2,994,550
**Types**	**DY-S1**		**DY-S2**		**DY-S3**	
	**Total**	**Unique**	**Total**	**Unique**	**Total**	**Unique**
Raw reads	13,215,515	2,349,112	15,702,355	2,967,602	18,147,568	3,540,661
3′adaptor & length filter	7,629,958	1,090,528	8,986,925	1,318,486	9,451,562	1,295,788
Junk reads	19,130	11,845	24,582	15,842	38,459	24,077
Clean reads	5,566,427	1,078,683	6,690,848	1,302,644	8,657,547	1,271,711
Rfam	1,204,287	19,546	1,339,068	21,195	1,608,522	23,011
mRNA	436,033	11,146	579,789	15,631	763,307	23,762
Repeats	19,530	245	25,234	273	21,092	273
valid reads	4,012,240	1,218,038	4,863,177	1,598,664	6,395,067	2,176,500

Note: 3′adaptor & length filter: reads removed due to 3′adaptor not found and length with <18 nt and >25 nt were removed. Junk reads: Junk: ≥2 N, ≥7 A, ≥8 C, ≥6 G, ≥7 T, ≥10 Dimer, ≥6 Trimer, or ≥5 Tetramer. Clean reads: equal to raw reads—3′adaptor & length filter—Junk reads. Rfam: collection of many common non-coding RNA families except microRNA, http://rfam.janelia.org, accessed on 5 September 2022. Repeats: prototypic sequences representing repetitive DNA from different eukaryotic species, http://www.girinst.org/repbase, accessed on 5 September 2022.

**Table 2 ijms-24-04069-t002:** miRNA–mRNA pairs showed the opposing expression under salt stress condition in DXWR.

miRNAs	Expression Trends	Target Genes	Expression Trends ^a^	Gene Name
bdi-miR5054_1ss10TA	up	Os01g0504100	down	*OsPUP8*
ath-miR8175_L-2	up	Os03g0130700	down	*-*
ath-miR8175_L-2_1ss20AT	up	Os03g0130700	down	*-*
gma-miR6300_1ss18GC	up	Os03g0219100	down	*-*
gma-MIR4995-p5_1ss18GC	up	Os03g0637900	down	*-*
ptc-MIR6476a-p3_2ss6AG18AC	up	Os04g0477000	down	*-*
bdi-miR5054_1ss10TA	up	Os05g0179300	down	*-*
gma-MIR6300-p5_1ss6AG	up	Os05g0219900	down	*-*
ath-miR8175_L-2	up	Os06g0495800	down	*-*
gma-miR6300_1ss18GC	up	Os07g0531500	down	*-*
gma-miR6300_R+1	up	Os07g0531500	down	*-*
bdi-miR5054_1ss10TA	up	Os08g0495500	down	*-*
osa-miR5072_L-4	up	Os10g0117000	down	*-*
ath-miR8175_L-1	up	Os10g0477900	down	*-*
ath-miR8175_L-2	up	Os10g0477900	down	*-*
ath-miR8175_L-2_1ss20AT	up	Os10g0477900	down	*-*
PC-5p-57749_50	up	Os10g0532200	down	*-*
osa-MIR1846e-p5_1ss15AG	up	Os11g0107700	down	*-*
gma-MIR4995-p5_1ss20GC	up	Os11g0170000	down	*-*
osa-MIR169g-p3	down	Os02g0596000	up	*-*
osa-MIR169h-p3	down	Os02g0596000	up	*-*
osa-MIR169j-p3	down	Os02g0596000	up	*-*
osa-MIR169l-p3	down	Os02g0596000	up	*-*
osa-MIR169m-p3	down	Os02g0596000	up	*-*
osa-MIR6251-p3	down	Os02g0756800	up	*-*
osa-MIR159a-p5	down	Os03g0130300	up	*DEFL8*
osa-miR3979-5p	down	Os03g0386500	up	*-*
osa-miR172d-5p_R+1	down	Os04g0398000	up	*OsERF101*
osa-miR827	down	Os04g0573000	up	*OsSPX-MFS1*
osa-miR399j_R-1	down	Os04g0691900	up	*-*
osa-miR5540	down	Os05g0582600	up	*OsSCP30*
osa-miR172d-5p_R+1	down	Os06g0154200	up	*D3*
osa-miR169r-3p	down	Os06g0569500	up	*OsKOS1*
osa-MIR159a-p5	down	Os07g0637300	up	*OsPDK1*
vvi-MIR3638-p5_2ss17GT18CT	down	Os08g0425800	up	*-*
osa-MIR812r-p3	down	Os10g0181200	up	*-*
osa-MIR164f-p3	down	Os11g0454300	up	*OsRab16A*
osa-MIR164f-p3	down	Os11g0673000	up	*-*
osa-miR444b.1	down	Os12g0116100	up	*-*
osa-MIR1860-p5	down	Os12g0174100	up	*-*
PC-5p-75382_31	down	Os12g0491800	up	*-*

Note: ^a^, the expression trends of the genes were derived from the previous study [20].

## Data Availability

The data presented in this study are available in the article or Appendix A.

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
