# Peer review of "Genome-Wide Analysis of microRNAs and Their Target Genes in Dongxiang Wild Rice (Oryza rufipogon Griff.) Responding to Salt Stress"

_ijms, 2023, doi:10.3390/ijms24044069_

Round 1

Reviewer 1 Report

The study describes possible mechanisms of stress tolerance in the world's northernmost wild rice variety. Using genome-wide sequencing of miRNAs and stem-loop qRT-PCR, the authors try to deduce the role of miRNA in stress tolerance; GO analysis revealed the possible involvement of several candidate miRNAs as salt-responsive elements involved in stress tolerance. Although these preliminary data are certainly promising, further qRT-PCR analyses under stress conditions would be needed to more convincingly show the direct link of individual miRNAs to specific stress conditions.

Otherwise, it is an interesting topic, scientifically sound, with a satisfactory number of results – some of them unnecessarily presented in the Supplementary data section. I have several major suggestions and some minor comments.

Specific comments:

Section 3.1 (Overview of sRNA library data set)   I find this whole section too descriptive and of really little informational value. This entire section should be rewritten and the authors should consider presenting the sequencing results (Table 1) in a simpler graphical form (e.g. bar plot). I would suggest the authors merge this section with the next section, 3.2.

Fig. 2. This volcano plot is not suitable as a standalone picture. Instead, I would recommend merging it with Fig. 1 and presenting the results earlier in the text.

p. 6…. osa-miR166j-5p, and PC-5p-16439_287 were downregulated, which largely coincided with the high-throughput sequencing data…   I find this statement very vague and not really consistent with what we can see in Fig. 3. Some of the expression levels match up nicely, but there are also quite large discrepancies. Be specific, try to discuss these results.

Fig. 3. Graphic quality of this figure is absolutely unsatisfactory. The legend can be displayed either above or below the picture, not going through the graphic. No statistical analysis is shown.

Section 3.4 (Prediction and function…)    The whole section is again very descriptive and the analysis of GO terms does not yield any conclusive results, this should be completely revised.
   Are some GO terms more significantly enriched than others?
   The results of Table S5 should be described in more detail, Table S5 can also be presented in a simplified form in the main results here. (another panel in Fig 4)
   Also, quite important results about the upregulation of possible target genes by candidate miRNAs (Fig. S4) should, in my opinion, be described in much more detail and the corresponding figure should be presented here. These genes can be listed in this section and the discussion should be more oriented toward how these genes are related and presenting a possible working model.

   Os11g0454300 is upregulated two orders of magnitude higher than the others. Any comments on this? Can you perform some in silico analysis (e.g. Genevestigator) of this and other candidate gene transcripts responding to stress conditions?

Minor points:

Abstract… “has highly resistant” – change e.g. to “is highly resistant”

p. 3. change the name of the Section (Results)

Fig. 4…  graphic quality of this picture is not sufficient, please increase all font sizes

Author Response

Dear reviewer,

On behalf of all the authors, I sincerely thank you for your comments concerning our manuscript entitled “Genome-wide analysis of microRNAs and their target genes in Dongxiang wild rice (Oryza rufipogon Griff.) responding to salt stress” (ijms-2155022). We have carefully studied the comments and have made the corrections and additions accordingly. We invite you to have another review and comment the revised version to see if it meets the criteria for publication. At the revised manuscript, all changes have been highlighted for ease of identification.

With respect and warm regards,

Fantao Zhang

The study describes possible mechanisms of stress tolerance in the world's northernmost wild rice variety. Using genome-wide sequencing of miRNAs and stem-loop qRT-PCR, the authors try to deduce the role of miRNA in stress tolerance; GO analysis revealed the possible involvement of several candidate miRNAs as salt-responsive elements involved in stress tolerance. Although these preliminary data are certainly promising, further qRT-PCR analyses under stress conditions would be needed to more convincingly show the direct link of individual miRNAs to specific stress conditions.

Otherwise, it is an interesting topic, scientifically sound, with a satisfactory number of results – some of them unnecessarily presented in the Supplementary data section. I have several major suggestions and some minor comments.

Specific comments:

Section 3.1 (Overview of sRNA library data set) I find this whole section too descriptive and of really little informational value. This entire section should be rewritten and the authors should consider presenting the sequencing results (Table 1) in a simpler graphical form (e.g. bar plot). I would suggest the authors merge this section with the next section, 3.2.

Response: Thanks for your comments. We rewrote this entire section to make the text more concise. Meanwhile, we tried to present the sequencing results in a graphical form, but it is hard to show the details in full detail. Given such consideration, we would like to keep Table 1 in our manuscript. Also, the two sections (3.1 and 3.2) focused on different points, it is hard to merge them into one section. Thus, we want to present these results in two sections.

Fig. 2. This volcano plot is not suitable as a standalone picture. Instead, I would recommend merging it with Fig. 1 and presenting the results earlier in the text.

Response: Thanks for your comments. We changed this volcano plot into a heat map to present the results of differentially expressed miRNAs more detailed. Please see Figure 2.

… …. osa-miR166j-5p, and PC-5p-16439_287 were downregulated, which largely coincided with the high-throughput sequencing data…   I find this statement very vague and not really consistent with what we can see in Fig. 3. Some of the expression levels match up nicely, but there are also quite large discrepancies. Be specific, try to discuss these results.

Response: Thanks for your comments. We rewrote this section and added more discussion on this topic in the Discussion.

Fig. 3. Graphic quality of this figure is absolutely unsatisfactory. The legend can be displayed either above or below the picture, not going through the graphic. No statistical analysis is shown.

Response: Thanks for your comments. We made a revision to this Figure.

Section 3.4 (Prediction and function…) The whole section is again very descriptive and the analysis of GO terms does not yield any conclusive results, this should be completely revised.

Response: Thanks for your comments. We rewrote the whole section and added more discussion on this topic in the Discussion.

   Are some GO terms more significantly enriched than others?

Response: Thanks for your comments. We added more discussion on this topic in the Discussion.

   The results of Table S5 should be described in more detail, Table S5 can also be presented in a simplified form in the main results here. (another panel in Fig 4)

Response: Thanks for your comments. We added Table 2 to present the results in more detail and added more discussion on this topic in the revised manuscript.

   Also, quite important results about the upregulation of possible target genes by candidate miRNAs (Fig. S4) should, in my opinion, be described in much more detail and the corresponding figure should be presented here. These genes can be listed in this section and the discussion should be more oriented toward how these genes are related and presenting a possible working model.

Response: Thanks for your comments. We added Figure 5 to present the results in the revised manuscript and added more discussion on this topic in the revised manuscript.

   Os11g0454300 is upregulated two orders of magnitude higher than the others. Any comments on this? Can you perform some in silico analysis (e.g. Genevestigator) of this and other candidate gene transcripts responding to stress conditions?

Response: Thanks for your comments. We added more discussion on this point in the revised manuscript. In the previous study, we analyzed the transcriptome profiles of DXWR under salt stress by mRNA transcriptome sequencing. In this study, the expression levels of the five genes were coincided with those in the mRNA transcriptome sequencing. Meanwhile, the expression levels of the five genes were verified by qRT-PCR approach. The expression level of OsRab16A (Os11g0454300) was the most significantly changed among the five genes, which was coincided with the result of transcriptome sequencing and further implies the role of this gene in conferring salt stress tolerance in plants.    

Minor points:

Abstract… “has highly resistant” – change e.g. to “is highly resistant”

Response: Thanks for your comments. We revised it in the manuscript.

  1. 3. change the name of the Section (Results)

Response: Thanks for your comments. We revised it in the manuscript.

Fig. 4…graphic quality of this picture is not sufficient, please increase all font sizes

Response: Thanks for your comments. All the pictures were adjusted to no less than 300 dpi resolution.

Reviewer 2 Report

Rice yield decreases when exposed to abiotic stresses, such as salinity, which is one of the most detrimental factors for rice production. Dongxiang wild rice (DXWR) is a progenitor of cultivated rice and has highly resistant to salt stress. Chen et al. performed miRNA sequencing to identify miRNAs and their putative target genes in response to salt stress in DXWR. It could provide the new information for improving salt stress tolerance in cultivated rice in the future. However, this study only identified known and novel miRNAs in DXWR under salt stress. It is the raw results and no further work on functions of miRNAs and their target genes.

Author Response

Dear reviewer,

On behalf of all the authors, I sincerely thank you for your kind comments concerning our manuscript entitled “Genome-wide analysis of microRNAs and their target genes in Dongxiang wild rice (Oryza rufipogon Griff.) responding to salt stress” (ijms-2155022). We have carefully studied the comments and have made the corrections and additions accordingly. We invite you to have another review and comment the revised version to see if it meets the criteria for publication. At the revised manuscript, all changes have been highlighted for ease of identification.

With respect and warm regards,

Fantao Zhang

Reviewer 2:

Rice yield decreases when exposed to abiotic stresses, such as salinity, which is one of the most detrimental factors for rice production. Dongxiang wild rice (DXWR) is a progenitor of cultivated rice and has highly resistant to salt stress. Chen et al. performed miRNA sequencing to identify miRNAs and their putative target genes in response to salt stress in DXWR. It could provide the new information for improving salt stress tolerance in cultivated rice in the future. However, this study only identified known and novel miRNAs in DXWR under salt stress. It is the raw results and no further work on functions of miRNAs and their target genes.

Response: Thanks for your kind comments. Dongxiang wild rice (Oryza rufipogon Griff., DXWR) is a cultivated rice progenitor that can survive in harsh environments, which makes it extremely valuable since it will allow us to study the mechanisms of abiotic stress resistance. However, the regulatory mechanism of miRNA-mediated salt stress-response in DXWR remains unclear. In this study, miRNA sequencing and bioinformatics analysis were performed to identify miRNAs and their putative target genes in response to salt stress in order to better understand the roles of miRNAs in DXWR salt stress-tolerance. These results provided an abundant resource of candidate miRNAs and target genes associated with salt stress-tolerance and enriched its regulatory network in plants. In the revised manuscript, we discussed the potential significance of these findings in more detail.

We think that the identified salt stress-responsive miRNAs from DXWR could provide a basis for developing salt stress-resistant rice varieties through molecular design breeding in the future. We also believe this study provides valuable information for the readers.

Reviewer 3 Report

In this paper, miRNAs of salt stress tolerant Dongxiang wild rice under normal and salt stress condition was studied. The following corrections are required in the manuscript

Abstract:

“progenitor of cultivated rice and has highly …..” should be changed to “progenitor of cultivated rice and has high tolerance …..”

In other places of this manuscript also use the word “tolerance/tolerant” in place of “resistance/resistant”. As it is an abiotic stress which is governed by multiple genes in a global regulation, thus resistance against tis abiotic stress is not possible.

Please mention endogenous control in the section 2.4

Considering your RT-qPCR data, how did you normalize your expression data using 1 or 2 reference genes? Please provide more details.

In table 1, clearly mention within the table that which data belong to control and which one for salt stress.

Towards end of the discussion, add one para on future prospects in term of commercial utility of the miRNAs in plant physiological responses under salinity.

Author Response

Dear reviewer,

On behalf of all the authors, I sincerely thank you for your kind comments concerning our manuscript entitled “Genome-wide analysis of microRNAs and their target genes in Dongxiang wild rice (Oryza rufipogon Griff.) responding to salt stress” (ijms-2155022). We have carefully studied the comments and have made the corrections and additions accordingly. We invite you to have another review and comment the revised version to see if it meets the criteria for publication. At the revised manuscript, all changes have been highlighted for ease of identification.

With respect and warm regards,

Fantao Zhang

Reviewer 3:

Comments and Suggestions for Authors

In this paper, miRNAs of salt stress tolerant Dongxiang wild rice under normal and salt stress condition was studied. The following corrections are required in the manuscript

Abstract: “progenitor of cultivated rice and has highly …..” should be changed to “progenitor of cultivated rice and has high tolerance …..”

Response: Thanks for your comments. We revised it in the manuscript.

In other places of this manuscript also use the word “tolerance/tolerant” in place of “resistance/resistant”. As it is an abiotic stress which is governed by multiple genes in a global regulation, thus resistance against tis abiotic stress is not possible.

Response: Thanks for your comments. We revised it in the whole manuscript.

Please mention endogenous control in the section 2.4

Response: Thanks for your comments. We added the endogenous controls in the section Materials and methods.

Considering your RT-qPCR data, how did you normalize your expression data using 1 or 2 reference genes? Please provide more details.

Response: Thanks for your comments. We provided more detailed descriptions in the revised manuscript. U6 small nuclear RNA (U6snRNA) and actin genes were used as endogenous controls to normalize the threshold cycle (Ct) values for miRNA and mRNAs qRT-PCR, respectively. The relative expression levels of the miRNAs and mRNAs were calculated using the 2−ΔΔCt method.

In table 1, clearly mention within the table that which data belong to control and which one for salt stress.

Response: We are sorry for the mistake. We revised it in Table 1.

 Towards end of the discussion, add one para on future prospects in term of commercial utility of the miRNAs in plant physiological responses under salinity.
Response: Thanks for your comments. We added more discussion on this topic in end of the Discussion.

Round 2

Reviewer 1 Report

In the revised version of their manuscript, the authors addressed most of the comments and suggestions and considerably improved the text’s quality. The text now shows much better focus while keeping sufficient depth and a range of interrelated biological processes centered around the role of miRNAs in stress tolerance are discussed. The revised manuscript appears to be acceptable for publication.

Minor comments:

Please update Fig. 5 legend. How are the comparisons made?

Author Response

Point-to-point response:

Please update Fig. 5 legend. How are the comparisons made?

Response: Thanks for your comments. We updated Fig. 5 legend. Please see below and the revised manuscript.

Figure 5. The expression levels of target genes under salt stress detected by qRT-PCR. The blue bars represent the samples under normal condition, and the orange bars represent the samples under salt stress. The relative expression was calculated by the 2-ΔΔCt method, and the standard deviation was calculated with three biological repeats.

Reviewer 2 Report

The response is fine and I think it could be accepted.

Author Response

We would like to thank you very much for your recognition of our work and valuable comments.